# Estimation of the Genetic Components of (Co)variance and Preliminary Genome-Wide Association Study for Reproductive Efficiency in Retinta Beef Cattle

**DOI:** 10.3390/ani13030501

**Published:** 2023-01-31

**Authors:** José María Jiménez, Rosa María Morales, Alberto Menéndez-Buxadera, Sebastián Demyda-Peyrás, Nora Laseca, Antonio Molina

**Affiliations:** 1CEAG The Council of Cadiz, 11400 Jerez de la Frontera, Spain; 2Department of Genetics, Veterinary School, Campus de Rabanales, University of Córdoba, Edificio Gregor Mendel, Ctra. Madrid-Cádiz, km 396, 14014 Córdoba, Spain; 3Departamento de Producción Animal, Facultad de Ciencias Veterinarias, Universidad Nacional de La Plata, La Plata 1900, Argentina; 4Consejo Nacional de Investigaciones Científicas y Técnicas (CONICET), La Plata 1900, Argentina

**Keywords:** Retinta beef cattle, age at first calving, calving interval, reproductive efficiency, random regression model, GWAS

## Abstract

**Simple Summary:**

Fertility is one of the most important traits for productivity in extensive beef production systems, since it has a major effect on the number of calves born and, as a result, the quantity of weaned calves produced per year. However, it is difficult to improve this trait under extensive conditions, mainly due to the lack of reliable and easy-to-obtain selection criteria. In this study, fertility was analyzed using reproductive efficiency, which was calculated as the deviation between the optimal and real parity number of females at each age. We demonstrated a high h^2^ value (0.30) using a classic repeatability model and a random regression model (ranging from 0.24 to 0.51), which suggests that the latter model can be recommended to improve fertility in beef breeds raised under extensive environmental conditions such as the Retinta. In addition, we performed the first GWAS analysis looking for SNP genetic markers associated with this character in cattle, which showed five markers significantly associated with the trait located on BTA4 and BTA28. Finally, the functional analysis revealed the presence of five candidate genes located within these regions, which were previously shown to be related to fertility in cattle and mice models.

**Abstract:**

In this study, we analyzed the variation of reproductive efficiency, estimated as the deviation between the optimal and real parity number of females at each stage of the cow’s life, in 12,554 cows belonging to the Retinta Spanish cattle breed, using classical repeatability and random regression models. The results of the analyses using repeatability model and the random regression model suggest that reproductive efficiency is not homogeneous throughout the cow’s life. The h^2^ estimate for this model was 0.30, while for the random regression model it increased across the parities, from 0.24 at the first calving to 0.51 at calving number 9. Additionally, we performed a preliminary genome-wide association study for this trait in a population of 252 Retinta cows genotyped using the Axiom Bovine Genotyping v3 Array. The results showed 5 SNPs significantly associated with reproductive efficiency, located in two genomic regions (BTA4 and BTA28). The functional analysis revealed the presence of 5 candidate genes located within these regions, which were previously involved in different aspects related to fertility in cattle and mice models. This new information could give us a better understanding of the genetic architecture of reproductive traits in this species, as well as allow us to accurately select more fertile cows.

## 1. Introduction

The productivity of extensive beef production systems relies on the sum of several traits. Of these, fertility is one of the most crucial, since it has a major effect on the number of calves born and, therefore, the quantity of weaned calves produced per year [1]. This has been demonstrated by several studies, which went so far as to quantify reproductive traits as three times more important than production traits in semi-intensive cattle production systems [2,3,4,5]. Despite the existence of a sizeable environmental effect, it was also demonstrated that there is a considerable genetic component affecting fertility in cows, thus enabling us to select more fertile individuals in order to increase the reproductive efficiency of the whole system [6]. In this context, there is a general consensus that the ideal beef cow, in terms of fertility, should be precocious (i.e., first calving at 24 months in Retinta) and produce one calf per year [7,8]. Therefore, the possibility of selecting individuals that have a genetic predisposition to meet such objectives could represent a key advantage from a breeding point of view.

An increase in the calving interval is one of the most common causes of culling in beef cattle [9]. However, this reproductive trait, closely associated with overall fertility, is characterized by low heritability and slow genetic progress in cattle [10]. For this reason, it is rarely included as a selection criterion in breeding programs of beef breeds raised under extensive environmental conditions. Instead, breeding programs focus more on increasing the live weight of the individuals, using growth traits such as weight and average daily weight gain, which correlate negatively with fertility [11]. However, the inclusion of reproductive traits, either individually or in combination through indexes, has been considered in certain breeds, although it is not yet a very widespread practice [12]. Cammack, et al. [10] summarized average values and heritabilities in thirteen different reproduction traits in beef cattle raised under extensive systems reported over 25 years, showing that most of them focused on the effects of the bull and precocity. However, few studies have looked into the individual fertility of cows, most likely due to the difficulty in obtaining a large and reliable dataset in extensive or grazing production systems.

During the last few years, the use of an indirect fertility criterion, including precocity and the calving interval, which can be estimated indirectly based on reproductive records, has been demonstrated as an interesting option for selecting more fertile females in extensively-bred livestock species [13,14]. This parameter, called reproductive efficiency (Re), is estimated as the percentage deviation of the number of calvings that a cow has at each age, from the number of calvings that this cow could have had in optimal conditions. In the Retinta breed, the optimum age at first calving was considered two years and one year for the optimal calving interval [15]. It has demonstrated increased heritability and reliability in comparison with other reproductive traits in beef cattle, such as calving interval or age at first calving [16]. However, to our knowledge, it has only been analyzed in a few studies in cattle [17,18]. In addition, reproductive traits have mostly been evaluated from a genetic point of view, using repeatability models. This methodology assumes that the Re values at the different calvings are manifestations of the same trait, and therefore the (co)variance across the trajectory of the parities of the cow is constant. Despite the fact that this methodology is widely used, Wilson and Réale [19] concluded that this assumption is not correct since the genetic parameters for this kind of effect vary at different ages. However, there is still a shortage of reports evaluating the changes in such parameters over the lifetime of the cows.

In recent years, the genomic revolution has allowed us to identify single nucleotide polymorphisms (SNPs) associated with phenotypic traits, leading to a better understanding of complex traits. One of the most powerful tools for analyzing such associations is the genome-wide association study (GWAS). In beef, a large number of GWAS analyses have been reported for different traits such as longitudinal, carcass, fatty acid profiles, meat tenderness and quality, and growth traits [20,21,22,23,24,25,26]. However, GWAS studies on reproductive traits in beef cows are still scarce [27,28,29,30].

Retinta is an autochthonous breed with unique adaptive characteristics linked to an extensive regime in the Dehesa ecosystem. Nowadays, it is widely bred in the south of Spain, due to its excellent adaptation to the harsh environment characterized by marginal pasturelands and the hot, dry climate [31]. In 2018, the breed reported more than 42,000 breeding cows, of which 16,850 were enrolled in the Genealogical Book [32]. Of these, 25% were raised as purebreds, with the rest used in crossbreeding as a maternal line with other continental breeds, such as Charolais and Limousin, with the aim of increasing profitability by using crossbred individuals [33]. For over 40 years now, the Retinta breeders’ association has not only gathered pedigree records but also a large phenotypic and environmental dataset, including birth and culling records from all the individuals enrolled in the breed [15]. For this reason, the Retinta breed is an interesting model by which to evaluate quantitative traits accurately, including those related to fertility.

In this study, we aimed to determine the genetic influence on the fertility of Retinta breeding cows by measuring the variance component patterns of reproductive efficiency across the calvings in the Retinta beef breed using random regression models (RRM). Additionally, we performed a preliminary genome-wide association study for reproductive efficiency traits to identify genetic variants, genomic regions, and candidate genes associated with fertility.

## 2. Material and Methods

### 2.1. Animal Dataset

In this study, we analyzed 63,421 calving records collected by the National Retinta Breeders’ Association. The dataset comprises information on 13,888 cows (from 1356 sires and 9154 dams, of which 5968 are in the data vector), which produced offspring with 3922 different sires. The pedigree was extended to include all the available information in the breed database, with a total of 20,178 animals. The inbreeding coefficients of the cow (Fc) and the bull (Fs) were determined according to the methodology described by Meuwissen and Luo [34] using the optiSel package [35] from the R statistical environment [36] and clustered into 15 classes of 2 percent intervals each. In addition, we estimated the calving number (Cn) and the herd-year-breeding season combination (HYS) in each observation using self-made R-scripts together with the Tidyverse [37] and data.table [38] packages. We estimated the fertility of the cows using the Re parameter, calculated as the deviation between the optimal and real parity number of females at each age, as described by Perdomo-Gonzalez, et al. [39]. After filtering and pruning incomplete and outlying data, 57,018 records from 12,554 cows were retained for the analysis.

### 2.2. Quantitative Genetic Analysis

Genetic analysis was performed using 2 animal models implemented in ASreml3 [40], without including Fs since it was not significant (*p* > 0.05). First, we tested the classical repeatability model (Rep), in which it is assumed that the Re effect and residual variance are homogeneous across the Cn scale, as follows:$$y_{\mathrm{ijkl}}\approx µ+X_{1}{\mathrm{Cn}}_{i}+X_{2}{\mathrm{Fc}}_{j}+{\;Z}_{1}a_{m}+Z_{2}p_{n}+Z_{3}{\mathrm{HYSN}}_{k}+e_{\mathrm{ijkmn}}\to \mathrm{Rep}$$
where
y is the dependent variable Re.µ is the average, Cn is the calving number (fixed effect), HYSN is the herd-year-season of birth of the cow combination (random effect), and Fc is the inbreeding value of the cow (fixed effect). X_1_, X_2_, Z_1_, Z_2_, and Z_3_ are incidence matrixes (0 or 1). e is the residual effect.

In addition, we tested a random regression methodology (RA), in which (co)variance components for Re varied across the Cn, assuming the existence of 6 classes of residual variance according to the calving number (e_res_ = 1,2,3,4,5, and 6 or more), using the following model:$$y_{\mathrm{ijkl}}\approx µ+\overset{n}{\underset{r=0}{\sum }}\Phi_{r}b_{1}{\mathrm{Cn}}_{i}+\overset{n}{\underset{r=0}{\sum }}\Phi_{r}b_{2}{\mathrm{Fc}}_{j}+Z_{3}{\mathrm{HYS}}_{k}+\overset{n}{\underset{r=0}{\sum }}\Phi_{r}a_{m:\mathrm{Cn}}+{\;I}_{\mathrm{Pn}}+e_{:\mathrm{res}}\to \mathrm{RA}$$
where b_1_ and b_2_ are the fixed regression coefficients modeled by a Legendre polynome (Φ) of order r = 2 representing the (co)variance variations for Re in Cn (9 classes) and Fc (16 classes), respectively. Z_3_ is an incidence matrix (0 or 1).

In both models, a_m:Cn_; p_n_ and HYS_k_ (herd-year-breeding season combination in each observation) are random effect vectors for the animal and their ancestors; and the repetition of records of the same trait in the animal and contemporary groups, respectively. Residual effects were considered homogenous ($e_{\mathrm{ijkmn}}$) in Rep, while they were estimated in 6 classes ($e_{:\mathrm{res}})\;$ according to Cn in RA. In Rep, $X_{1},\;X_{2},\;Z_{1},\;Z_{2}$, and $Z_{3}$ are incidence matrixes (0 or 1) connecting fixed and random effects, which were replaced (with the exception of $Z_{3}$) by Legendre polynomial coefficients (Φ*)* of r = n order to estimate the results in a longitudinal way along the calving trajectory of the cows in RA. The parameters by RA model were estimated using two different approaches with r = 1 and r = 2.

The variance components were estimated per model as follows:
Rep model
$$\mathrm{Var}\left(y\right)={A\sigma }_{a}^{2}+I_{p}\sigma_{p}^{2}+I_{\mathrm{HYS}}\sigma_{\mathrm{HYS}}^{2}+I_{n}\sigma_{e}^{2}$$RA model
$$\mathrm{ar}\left(y\right)=N[0,(\sigma_{y}^{2}=\left[G_{0}=\left(A⊗K_{a}\right)\right]+I_{p}\sigma_{p}^{2}+I_{\mathrm{HYS}}\sigma_{\mathrm{HYS}}^{2}+I_{n}\sigma_{e:\mathrm{res}}^{2}$$

In both cases, A is the additive genetic relationship matrix; I is the identity matrix (number of cows) of the p order. In Rep, $\sigma_{a}^{2}$, $\sigma_{p}^{2}$, $\sigma_{\mathrm{HYS}}^{2},$ and $\sigma_{e}^{2}$ are the additive, permanent environmental, HYS and residual variances, respectively. The genetic parameters and the animal Expected Genetic Values (EGV) for Re were estimated through the direct solution of the Rep model, assuming that they do not vary across Cn.

In the RA model, the same parameters were estimated along the Cn, using the K_a_ matrix
$$K_{a}=\Phi_{i}\left[\begin{array}{ccc} \sigma_{a_{o}}^{2} & \sigma_{a_{\mathrm{os}}} & \sigma_{a_{\mathrm{oq}}}\\ \sigma_{a_{\mathrm{so}}} & \sigma_{a_{s}}^{2} & \sigma_{a_{\mathrm{sq}}}\\ \sigma_{a_{\mathrm{qo}}} & \sigma_{a_{\mathrm{qs}}} & \sigma_{a_{q}}^{2} \end{array}\right]\Phi_{i}^{\prime }\;$$
containing the elements related to the intercept, the slope, and the quadratic term for the additive genetic effects with variances $\sigma_{a_{o}}^{2},$ $\sigma_{a_{s}}^{2}$ y $\sigma_{a_{q}}^{2}$ and their respective covariances $\sigma_{a_{\mathrm{so}}}$; $\sigma_{a_{\mathrm{sq}}}$ y $\sigma_{a_{\mathrm{oq}}}$; and $I_{p}\sigma_{p}^{2}+I_{\mathrm{HYS}}\sigma_{\mathrm{HYS}}^{2}+I_{n}\sigma_{e}^{2}$ represent the same indicators for the variances and covariances.

(Co)variance components were estimated along the Cn trajectory following the methodology proposed by Jamrozik, et al. [41], using the formula $\sigma_{a_{i}}^{2}=\Phi_{i}K_{a}\Phi_{i}^{\prime }$ for both genetic variances and covariances.

Heritability (h^2^) and repeatability (r) were estimated in Rep by the classical procedure proposed by Falconer and MCKay [42], while h^2^ and genetic correlations (r_g_) were estimated in each ith Cn point included in the respective Legendre polynomials ($\Phi_{i}$) in the RA model as follows:$$h_{i}^{2}=\frac{\Phi_{i}K_{a}\Phi_{i}^{\prime }}{\Phi_{i}K_{a}\Phi_{i}^{\prime }+I_{p}\sigma_{p}^{2}+I_{\mathrm{HYS}}\sigma_{\mathrm{HYS}}^{2}+I_{n}\sigma_{e:\mathrm{res}}^{2}}\;y\;r_{g_{\mathrm{ij}}}=\frac{\Phi_{i}K_{a}\Phi_{j}^{\prime }}{\sqrt{\Phi_{i}K_{a}\Phi_{i}^{\prime }{*\Phi }_{j}K_{a}\Phi_{j}^{\prime }}}$$

In addition, the RA model allows us to determine the EGV per individual (a_m_) on all the ith points of the Cn trajectory as follows:$${\mathrm{EGV}}_{m}^{i}=\Phi_{i}a_{m}^{\prime }$$
where $\Phi_{i}$ are the polynomial coefficients of the i point across the Cn trajectory, and a_m_ = [a_o_ a_s_ a_q_ ] is the animal genetic function estimated by the model described above.

Finally, the Re genetic values obtained using RA in each of the 9 calvings were summarized in a subjacent index (Ipc) estimated using a PCA analysis, according to the procedure described by Menendez-Buxadera, et al. [43].

In the second step, we arrived at a new EGV value based on the PCA results (IpcT) by applying the methodology described by Togashi and Lin [44], as follows:$${\mathrm{Ipc}}_{1}=\overset{9}{\underset{i=1}{\sum }}\mathrm{PC}1\mathrm{evi}*\mathrm{EGVi}\;{\mathrm{Ipc}}_{2}=\overset{9}{\underset{i=1}{\sum }}\mathrm{PC}2\mathrm{evi}*\mathrm{EGVi}$$
IpcT = Ipc_1_ + Ipc_2_
where EGV_i_ are the genetic values of the ith calving, expressed as standardized values, and PC1ev_i_ and PC2ev_i_ are the estimations of the PC1 and PC2 for the ith calving, respectively.

### 2.3. DNA Samples and Genotyping of Samples

In total, 252 Retinta cows were selected for genotyping by their EGV index based on the PCA results (IpcT), applying the methodology described by Togashi and Lin [44] (described in the Quantitative Genetic Analysis section). Of these, 40 cows had a negative IpcT, and 212 cows had an IpcT in the upper quartile. Genomic DNA was isolated from blood samples using a DNeasy Blood & Tissue Extraction Kit (Qiagen, Hilden, Germany). The cows were genotyped using the Axiom Bovine Genotyping v3 Array, which includes 63,000 markers uniformly distributed across the genome. The raw genotype data were processed following the ‘‘Best Practices Workflow” procedure in the Axiom Analysis Suite package v5.0 and were filtered using PLINK v1.9 software [45]. SNP markers with a call-rate ≤ 0.95 and with a minor allele frequency (MAF) < 0.05 were excluded. In the final step, the dataset was pruned by linkage disequilibrium in PLINK using the *–indep-pairwise 50 5 0.5* command. The final genomic dataset included 29,037 SNPs from the autosomal and X chromosomes.

### 2.4. Genome-Wide Association Study (GWAS)

GWAS analysis was performed using GEMMA software [46], employing the following univariate linear mixed model:y = Wα + Xβ +µ + ɛ 
where y is an n-vector with pseudo-phenotypes (IpcT index); W is an incidence matrix of covariates (fixed effects) including a column of 1s; α is a vector of the corresponding coefficients including the intercept; X is an n-vector of marker genotypes; β is the effect size of the marker; µ is an n-vector of random effects µ ~ N(0, λτ ^−1^ K), where τ^−1^ is the variance of the residual errors; λ is the ratio between two variance components; K is the genomic relationship matrix (estimated from the markers); and ɛ is the n-vector of errors. In addition, the model included a correction for population stratification based on the first 10 components of a principal component analysis performed in PLINK v.1.9 as covariates. The statistical significance of the SNP effect was calculated using the Wald test statistic, which determined a *p*-value for each SNP.

### 2.5. Functional Analysis

Potential candidate genes associated within ± 500 Kb of the significant SNPs were annotated using the Ensembl BioMart database with the last available cow reference genome (ARS-UCD1.2. https://www.ensembl.org/Bos_taurus/Info/Index (accessed on 28 September 2022). Finally, the function of these genes and their putative relationship with fertility processes was established by performing an extensive review of the available literature in public databases, as well as in the DAVID V6.8 and Uniprot online resources.

## 3. Results and Discussion

Fertility is one of the key factors in any extensive livestock production system. However, at the same time, it is one of the most neglected from a genetic point of view, since it is difficult to study models in such production conditions. For this reason, the use of indirect traits obtained from pedigree records is becoming an interesting alternative to evaluating fertility at the population level since it allows us to obtain large phenotypic datasets, which are crucial for making reliable estimations [13,47]. Using such an approach, we were able to determine the genetic and environmental effects in a large cohort of beef cows bred in extensive conditions. To our knowledge, no previous studies have employed this methodology in cattle.

Re is a novel trait that estimates the deviation between the optimal and real parity number of females at each age of the cows (considering precocity and calving interval) throughout their entire productive lives. Our results showed a wide variability in average Re (across calving) among Retinta cows (Table 1).

Furthermore, Re increased in value almost linearly with the calving number (Figure 1). This positive correlation (0.97) agrees with the results obtained by Mercadante, et al. [48], and Swanepoel and Hoogenboezem [49], who concluded that the cows that remain longer in the herd present better reproductive behavior during their productive lives. In this context, Balieiro, et al. [50] demonstrated that the most fertile cows that do not “miss” any breeding seasons are the ones most likely to be allowed to remain in the herd in extensively-raised beef breeds. Interestingly, those “regular and reliable” individuals show high values of Re, demonstrating the validity of this trait as an indirect estimator of fertility.

The same analysis revealed lower variability in the age of first calving (33.9 ± 6.18 months, on average), which can be associated with differences in precocity, as has been convincingly proven in beef breeds [51,52]. This fact is particularly important since precocity is negatively related to longevity, and cows starting their reproductive lives at later stages are less likely to be culled for reproductive reasons [53]. Since precocity is not commonly included in models of calving intervals in beef cattle, we hypothesize that Re may be an interesting estimator of fertility in cows bred in extensive conditions, reducing a potential cause of bias within the evaluations.

### 3.1. Modeling the Genetic Influence on Re Using a Repeatability Model

In this study, we employed two different approaches to estimate the genetic component affecting Re: the classical Rep model and the RRM. The former is the most common methodology employed, having been used for over 20 years [16]. However, the latter has been increasingly employed during recent years since it allows us to estimate changes in the variance components of Re (or any reproductive trait) across calvings and, therefore, to select individuals with increased fertility values at any particular age [54]. However, regardless of which model is employed, fertility traits in livestock bred in extensive conditions must be analyzed using large reproductive datasets, such as the dataset used with Re as the trait of interest.

The variance components obtained using the Rep model are shown in Table 2. Heritability was particularly high (near 0.3) for a reproductive character in comparison with previous reports on beef cows, which are closer to 0.1 (reviewed by Cammack, et al. [10]). However, none of these employed Re as a fertility estimator for the cows. Likewise, similar h^2^ values (near 0.25) were recently reported in goats and horses [13,47]. In addition, it was also noticeable that the permanent environmental effect of the cow accounted for nearly 35% of the total variance and the contemporary group (HYS) for only 9%. This result for the permanent environmental effect is expected in any extensive system in which environmental factors (such as metritis or retained placentas) with poor treatment can permanently delay the cow’s fertility within the season [55]. Regarding HYS, the lower value fits in with the homogeneity of the production conditions across herds and years in the Spanish Dehesa in which the Retinta cows are bred [33].

Figure 2 shows the evolution of Re over the years under study, where a clear positive trend in the breeding values can be seen over recent decades. This could be partly due to selection against carriers of the Robertsonian translocation 1/29 (rob(1;29)). Despite the incidence of carrier individuals in the whole population decresing during the last 30 years from 15.73% in 1992 to 1.06% in 2020, an increase was observed in Re phenotypic values in the same period. Although this increase in Re can be partially related to the decrease in the translocation rate, it may also be explained partly by other effects, such as improvements in animal management [15].

### 3.2. Modeling Fertility across the Lifespan of the Cow

One of the drawbacks of the Rep model is the assumption of homogeneity of the variance components of Re across calvings. This hypothesis can be problematic for traits with low-moderate heritability, such as fertility-related traits. In contrast, RA models take into account the possible h^2^ variations across the stage of production of the individual (the calving number of the cow, in our case) [54]. In our study, both RA models showed better adjustments than those obtained using the Rep model (lower logL, AIC, and BIC, Table 3), which supports the hypothesis that Re behavior is not homogeneous for each calving of the cow. Among the RA models, the second-order RA (RA2) showed the best adjustment to the data, suggesting that it can produce the most unbiased estimation of the (co)variance components.

In the RA model, heritabilities rose gradually with increased calving number, from 0.246 in the 1st calving to 0.583 in the 9th, mostly due to a slower pace of reduction observed in genetic variance in comparison with phenotypic variance (Table 4). In this context, only the estimated heritability at the first calving in RA was lower than the estimated value using the Rep model. Therefore, the use of RA models allows us to give a better explanation of the intrinsic variation of the trait, which is not possible using Re. However, the reduction in the total variance observed for the later calvings can limit the selection possibilities of the trait, despite the higher heritabilities observed. This can be seen clearly in Figure 3, which shows the range of Re variation per calving. The size of this variation is close to 40% for the first calving but falls to 29% and 24% in the fourth and rest of the calvings, respectively, which implies that a better response to the selection based on EGVs is expected in the first calving. However, it is also worth mentioning that the correlated responses for the rest of the calvings will be lower, due to the decreasing pattern observed in the genetic correlations between the first and other calvings (Figure 4), despite the fact that all the correlations estimated among the genetic values for different calvings were high (none were lower than 0.6, Figure 4), and even higher (>0.83) when three or more calvings were considered. For this reason, our results suggest that the Re estimation for the third calving could be the best trait for selecting cows for fertility since it combines precocity in the selection practices, genetic variability, and reliability. In addition, it demonstrates that the order of merit for Re will vary among calvings, and therefore, selective practices should take this into account.

Finally, the dataset employed in this study allows us to analyze the evolution of EGV for Re during the last 3 decades, since the beginning of the improvement program (from 1992 to 2018, Figure 5), which showed a positive trend towards increased Re genetic values in both sexes (7% on average). However, this value was higher in the EGV of the 1st and 3rd calving, which implies that breeding decisions based on fertility were made by breeders at the early stages of the lives of these cows and bulls.

### 3.3. Practical Implications of the RA Results

One of the problems of RA models is the vast amount of information produced in each analysis. In our case, we estimated 9 EGVs for 38,058 individuals, making it difficult to obtain reliable and clear conclusions. To cope with this situation, we performed a multivariate principal component analysis (PCA) in which all the available information was transformed into vectors (principal components, PC), which account for a certain % of the (co)variances existing among the variables studied. In this procedure, originally developed by Togashi and Lin [44], PCs are ordered by the percentage of the total variance explained across all the response surfaces (all the calvings analyzed, in our case), in order to summarize the information produced by the RA analysis in a simple and useful way.

Our analysis revealed that 99.7% of EGV (co)variance was explained by two major $\lambda_{i}\;$(PC1 and PC2), which demonstrates that all the variability determined by the RA analysis can be interpreted in a reliable way using this methodology (Table 5). In our case, all the eigenvectors (EVs) in PC1 showed a very similar positive trend (close to 0.3), suggesting that any variation observed will be linear across the calvings (size vector). In contrast, PC2 (known as the vector of shape) showed positive values until the 4th calving, when it became negative.

Finally, the estimated IpcT (calculated as the combination of Ipc values for PC1 and 2) showed a much more accurate fit to the breeding value distribution observed in all the population in comparison to that obtained by estimating the average EGV values for each individual using the RA model (Figure 6). This value has an average of 0 ± 0.014 varying from −6.812 to 9.70. However, IpcT also allows us to summarize in a single value all the Re variability per individual across the calvings, thus allowing us to increase the accuracy of the breeding decisions. The correlation between the breeding values for Rep and IpcT shows a significant relationship of medium magnitude (0.45), which supports the use of this index as a pseudophenotype instead of the global EGV of the animal for this trait.

In addition, the use of IpcT permits a significant reduction in the risk of the possible interaction genotype x environment interaction problems (a different genetic potential in each calving number, as shown the Table 5) by adding the eingvectors, which by definition are uncorrelated. The use of PCA in different species and breed selection [43,56,57,58] and the results of the correlations between the EGV of each calving and the IpcT demonstrate it (Table 6).

### 3.4. Differences among Individuals in Re Development and Variation in Calvings

One interesting finding is the fact that individuals with similar EGV values at early ages can show opposite behaviors in terms of fertility at later stages. This pattern is observed in Figure 7A, in which 100 individuals with the same genetic estimation for Re at the first calving diverge profoundly at the fifth calving, in which some individuals showed a positive EGV while others showed the opposite. In the same way, Figure 7B showed the Re estimation per calving of 100 individuals with the same value at the 5th calving, in which some of them showed a highly positive estimation at the 1st calving (with values ranging around 20%), whereas another group of individuals showed negative values at the same calving. From a biological point of view, the differences observed among individuals can be explained by the phenomenon known as “plasticity”, described by de Jong and Bijma [59] and largely employed in modeling production traits in livestock systems [60,61]. However, it is worth mentioning that 82% of the best 100 individuals ranked using EGV1 and IpcT were coincident, and the divergent behaviors were shown by less of the population, which supports the use of PCA analysis as a robust methodology to perform better matings.

In our case, this parameter, defined as the difference between the genetic values obtained in the same character in the 1st and 9th calving, showed wide variability over the whole population. In addition, we determined an asymmetrical distribution with an increased proportion of negative values, which suggests increased reproductive efficiency at the first calving. We hypothesize that this deviation could be caused by the increased environmental variability affecting the Re values at the first calving caused by the age of the cow, since the age at first calving is influenced not only by the age at puberty but also by when each breeder decides to include the heifer in the mating lot. This breeder’s influence is reduced when the calving number increases (in addition to reducing the variability of the parameter with the elimination of less fertile animals).

### 3.5. Genome-Wide Association Study (GWAS) and Functional Analysis

Genome-wide association analyses detected 5 SNPs significantly associated with reproductive efficiency (*p*-value < 10^−4^, Table 7), located in 2 genomic regions on chromosomes BTA4 and BTA28 (Figure 8). The estimated average effect of the significant SNPs in absolute values was 4.91. The in-silico functional analysis revealed the presence of 5 candidate genes (*NRF1, SSMEM1, CPA5, RYR2,* and *ZP4*) previously linked with known biological processes, molecular functions, and pathways related to fertility located within ± 500 Kb of the significant SNPs (Table 7), among which there were processes related to steroidogenesis, embryonic development, and spermatogenesis in cattle, goat, rabbit, mouse, and human models.

The candidate genes located within the significant regions of the SNP AX-124382279 have previously been described in important biological pathways. One of these genes was nuclear respiratory factor 1 (*NRF1*), which is involved in mitochondrial biogenesis, signal transduction, and protein synthesis. In mice, this gene has been related to late gestational embryonic lethality when a loss of function occurs [62]. Recently, Zhang, et al. [63] demonstrated the effects of *NRF1* on steroidogenesis and cell apoptosis in goat luteinized granulosa cells. An attenuated expression of *NRF1* led to mitochondrial dysfunction, disrupted the cellular redox balance, impaired steroid synthesis, and finally resulted in granulosa cell apoptosis through the mitochondria-dependent pathway. The other two candidate genes found in this region were *SSMEM1* and *CPA5*, both related to male fertility. Serine-rich single-pass membrane protein 1 (*SSMEM1*) is a conserved testis-specific gene in mammals. Nozawa, et al. [64] demonstrated that *SSMEM1* is essential for male fertility in mice and found that the *SSMEM1* protein is expressed during spermatogenesis but not in mature sperm. The sterility of the *SSMEM1* KO mice was associated with globozoospermia and a loss of sperm motility, which is crucial for fertilization. Similarly, *CPA5* (carboxypeptidase A5) is involved in spermiogenesis and in the regulation of sperm function, as it is exclusively or highly expressed in spermatogenetic cells, especially from the secondary spermatocyte to elongated spermatid stages. However, Zhou, et al. [65] reported that there were no perceptible changes in reproductive phenotypes in *CPA5*-KO mice, suggesting that it is a dispensable factor for spermatogenesis and male fertility in mice.

On BTA28, we found the *RYR2* and *ZP4* genes located very close to the significant AX-169372743 marker. *RYR2* encodes an intracellular calcium-release channel and is expressed in many tissues, including the ovaries. In humans, some variants of *RYR2* were significantly associated with weight loss in early pregnancy [66], providing evidence that it may play a role in fertility. However, its expression was also associated with amphiregulin, a key mediator of the effect of LH/hCG and a marker for oocyte competence [67]. Finally, the *ZP4* gene codified a glycoprotein that formed the zona pellucida (ZP), the extracellular matrix that houses mammalian oocytes and embryos. In rabbits, Lamas-Toranzo, et al. [68] reported that *ZP4* has a structural role in the zona pellucida and that rabbits without *ZP4* were subfertile. In vitro, the loss of *ZP4* did not affect ovulation, fertilization, or the early stages of the development of embryos. However, in vivo development was severely impaired in embryos covered by a ZP4-devoid zone. In addition, two recent studies in rats showed that *ZP4* is not responsible for gamete-specific interaction [69,70].

## 4. Conclusions

In this study, we were able to model the fertility of cows bred in extensive conditions using a novel fertility trait, reproductive efficiency (Re), which can be estimated through the analysis of reproductive records, using two different genetic methodologies. Our results showed that Re has a significant genetic component, but also that the genetic influence on Re is not homogeneous during the length of life of the cows, and individuals were observed showing very different patterns of variability in the breeding values across parities, which cannot be detected by the Rep model. Therefore, the use of the RA model would be an option to enhance fertility in the Retinta breeding program. We also determined that the use of a composite index, based on a principal component analysis could satisfactorily integrate all the information produced by RA models in an efficient and user-friendly way for the breeding program. Finally, we demonstrated that evaluating reproductive efficiency using the RA model makes it possible to differentiate the cows’ genetic potential throughout the calving trajectory and enjoy greater flexibility in selecting female breeders. Finally, our preliminary GWAS analysis shows the existence of specific genomic regions influencing the fertility of Retinta cows. This new information could help us understand the genetic architecture of reproductive traits in the species better as well as allow us to select more fertile cows with greater accuracy. However, further analyses including large populations and different breeds would be required to validate our genomic findings.

## Figures and Tables

**Figure 1 animals-13-00501-f001:**
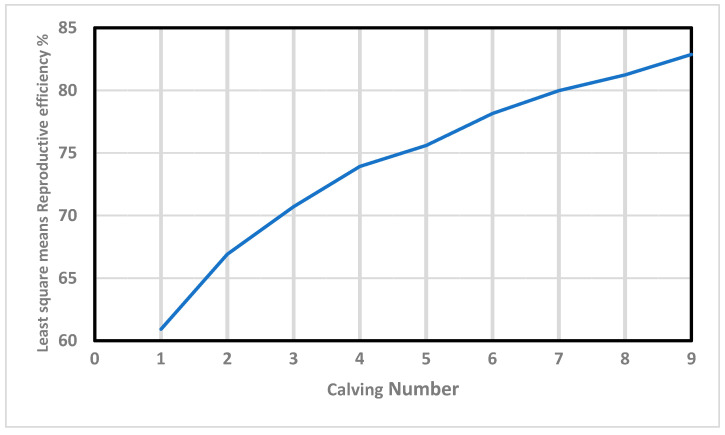
Effects of calving number on reproductive efficiency in Retinta breed.

**Figure 2 animals-13-00501-f002:**
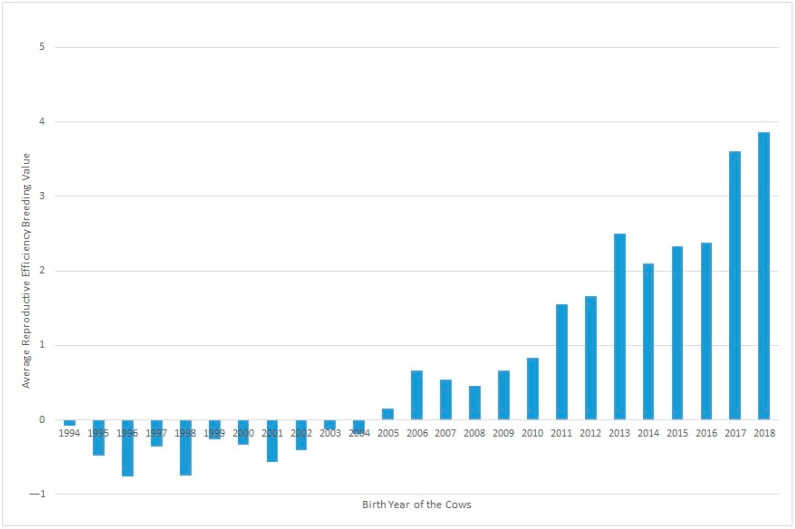
Evolution of the effect of the year of birth on the reproductive efficiency of the cow.

**Figure 3 animals-13-00501-f003:**
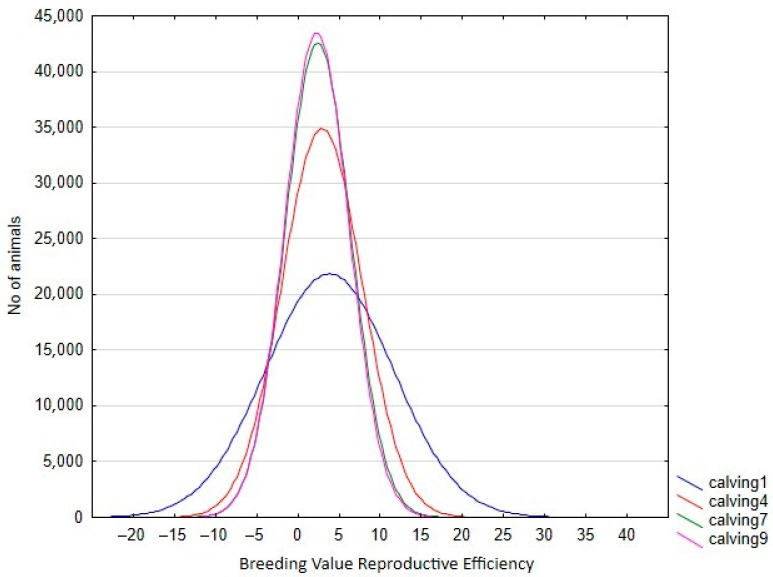
Variation in breeding value for reproductive efficiency per calving in the Retinta breed, estimated by random regression models.

**Figure 4 animals-13-00501-f004:**
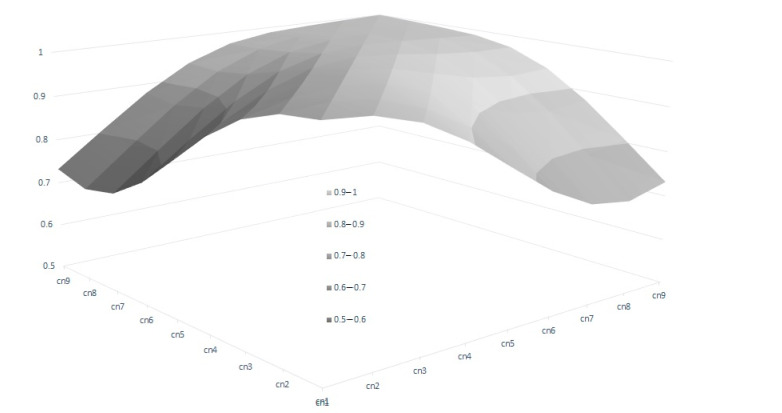
Genetic correlations of breeding value for reproductive efficiency between calvings, estimated by random regression genetic models.

**Figure 5 animals-13-00501-f005:**
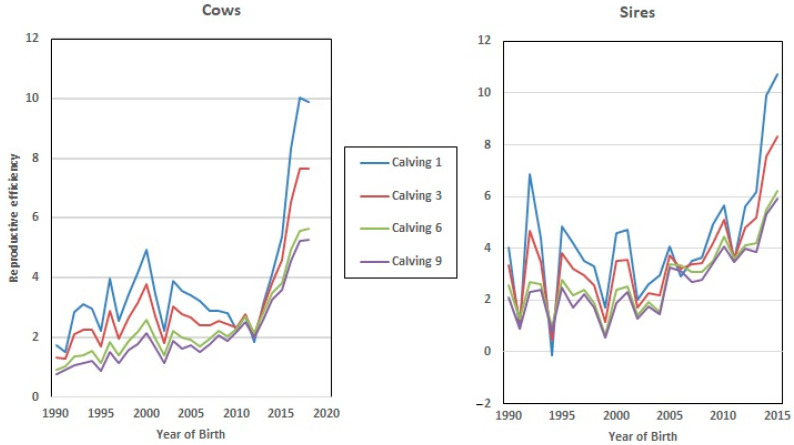
Genetic progress achieved in reproductive efficiency (Re) over a 30-year period.

**Figure 6 animals-13-00501-f006:**
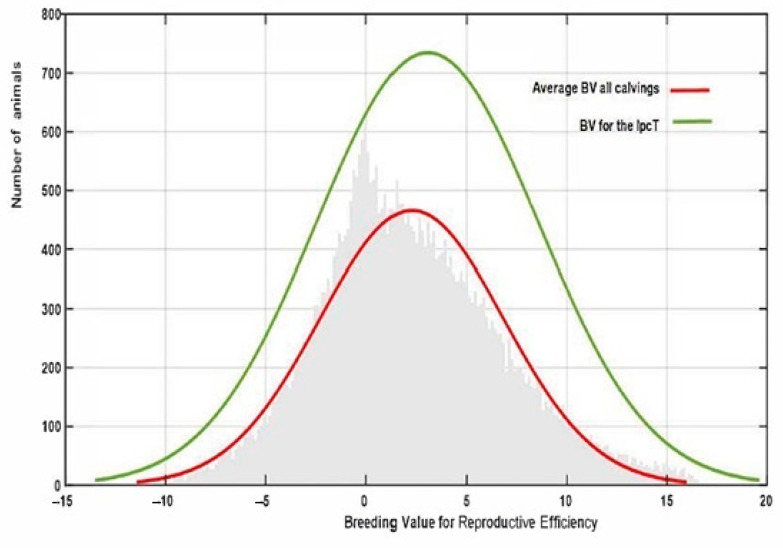
Variation in reproductive efficiency in the whole population estimated using PCA (IpcT) or the average RE_EGV_ for all the calvings.

**Figure 7 animals-13-00501-f007:**
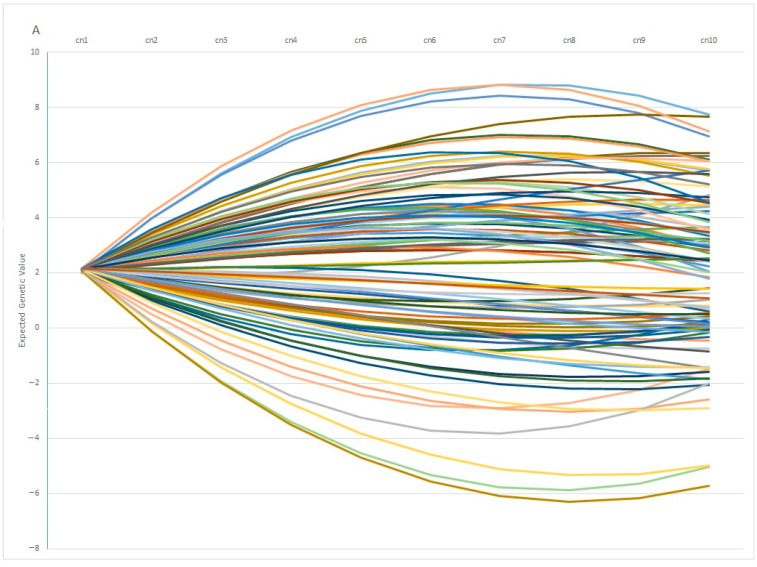
Variation on the Expected Genetic Values (EGV) across calvings (cn) in 100 individuals with similar EGV at 1st (**A**) or 5th calving (**B**).

**Figure 8 animals-13-00501-f008:**
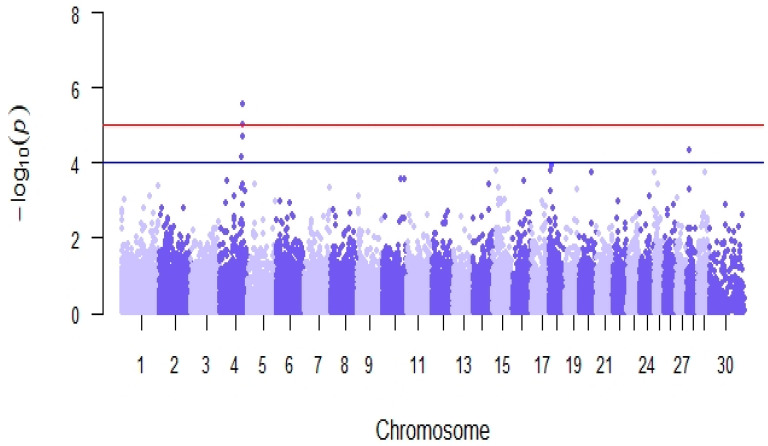
Manhattan plot of genome-wide association analyses for reproductive efficiency in the Retinta breed. The red line shows a genome-wide significance threshold (−log10(*p*) = 5) and the blue line a genome-wide suggestive threshold (−log10(*p*) = 4).

**Table 1 animals-13-00501-t001:** Descriptive statistics for input data.

Parameter	Mean	Minimum	Maximum	Coef.Var.
Reproductive efficiency (%)	72.05 ± 17.33	33	100	24.05
Inbreeding coefficient of the cow (Fc)	0.06 ± 0.08	0	0.46	136.00
Inbreeding coefficient of the bull (Fs)	0.08 ± 0.07	0	0.45	90.93
1st calving age (months)	33.99 ± 6.18	20	48	18.18

**Table 2 animals-13-00501-t002:** Variance components and parameter estimates for Reproductive efficiency.

Effect	Variance Components	% Relative Importance
Additive genetic effect	91.82	30.18
Permanent environmental effect	105.55	34.69
Herd-year-season effect	27.38	9.00
Residual effect	79.52	26.13
Total	304.27	

**Table 3 animals-13-00501-t003:** Comparison among repeatability (REP) and random regression models (RA).

Model	N of Parameters	logL	AIC	BIC
REP	4	−183,253	366,514	366,550
RA r = 1	11	−173,721	347,464	347,563
RA r = 2	14	−169,279	338,587	338,714

**Table 4 animals-13-00501-t004:** Genetic parameters for reproductive efficiency of the Retinta breed throughout the different calvings were estimated by repeatability (REP) and random regression models (RA).

	Random Regression Models of Second-Order		
Calving	Genetic Var	Total Var	h^2^
1	123.18	506.83	0.24
2	85.02	241.00	0.35
3	62.21	163.82	0.38
4	49.64	128.36	0.39
5	43.28	109.33	0.40
6	40.22	94.96	0.43
7	38.60	87.35	0.44
8	37.69	79.86	0.47
9	37.83	74.39	0.51
Repeatability model	91.82	304.27	0.30 ± 0.003

**Table 5 animals-13-00501-t005:** Estimated eigenvalues ($\lambda_{i}$) and eigenvectors (evi) for the principal component analysis results with the genetic values of reproductive efficiency for each calving, from a total of 38,058 animals.

Variable	PC1	PC2
ev1	0.29	0.58
ev2	0.31	0.45
ev3	0.32	0.29
ev4	0.33	0.11
ev5	0.33	−0.07
ev6	0.32	−0.21
ev7	0.32	−0.30
ev8	0.32	−0.33
ev9	0.32	−0.23
$\lambda_{i}$	9.37	0.59
Variance %	93.6	6.1

**Table 6 animals-13-00501-t006:** Correlations between the estimated EGV1 and the estimated EGV for each calving number and Correlations between the IpcT and the estimated EGV (first 9 calvings only).

	EGV1	EGV2	EGV3	EGV4	EGV5	EGV6	EGV7	EGV8	EGV9
EGV1	1.00	0.99	0.97	0.92	0.85	0.78	0.74	0.72	0.74
IpcT	0.97	0.99	0.99	0.99	0.96	0.92	0.90	0.89	0.91

**Table 7 animals-13-00501-t007:** List of SNPs associated with reproductive efficiency and candidate genes related to fertility in Retinta beef cattle.

Chr	SNP	Position (bp)	β	se	*p*-Wald	Candidate Gene
4	AX-124382279	93,739,544	−5.414	1.334	7.199054 × 10^−5^	*NRF1, SSMEM1; CPA5*
4	AX-106723907	97,052,195	−4.382	0.964	9.616885 × 10^−6^	
4	AX-185120444	97,052,759	−4.290	0.979	1.926352 × 10^−5^	
4	AX-115113656	97,128,968	−5.936	1.229	2.755483 × 10^−6^	
28	AX-169372743	10,126,454	−4.530	1.086	4.607665 × 10^−5^	*RYR2; ZP4*

## Data Availability

The data sets employed in this study are property of the ACRE (Asociación Nacional de Criadores de Ganado Vacuno Selecto de Raza Retinta—National Association of Breeders of Selected Retinta Cattles) and were provided for scientific purposes under a specific collaboration arrangement. The data set can be made available for scientific purposes to other authors by the ACRE technical department, upon reasonable request.

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
