# Peer review of "Estimation of the Genetic Components of (Co)variance and Preliminary Genome-Wide Association Study for Reproductive Efficiency in Retinta Beef Cattle"

_animals, 2023, doi:10.3390/ani13030501_

Round 1

Reviewer 1 Report

The subject of the paper is very interesting. I suggest some changes in order to better explain the analysis of the data and improve the results presentation. Minor changes needed are highlighted in the pdf attached. 

First, I suggest to revise the title, which is too long and confusing. 

Methodology: 

Line 129-130 - Is Herd-year-season related to the birth of the cow?

Is it possible to include the year/season of each breeding season in each observations? As the environment can be very different from one breeding season to another and it can affects the reproductive results. 

Lines 131 to 134 - I looked at the reference and did not understand clearly the calculation of the RE. As this is essential for the paper, I suggest to add a clear explanation of how the calculation was done. I suggest to add description of the distribution of the data, what is the mean, median, etc... Seems to me that the numbers varies from 40% to 100%. Therefore, it is a censored data as cannot be higher than 100%. Moreover, I think is important to explain how many animals with information in each calving number. How the cow culling process affects the estimation of the breeding values? 

Line 199 - As IpcT comes from a summ from several estimatives, is it changing the distribution of the data? or even the mean, median, maximum and minimum? I suggest to add this descriptive statistics in results section. 

Line 225 - 227 - Did you test not using the PCA correction? Because as you selected two contrasting groups of animals to compose the dataset, maybe you are loosing information of snps asssociated with the trait due this correction for population stratification. 

What breeding value was used for GWAS? Define it clearly. 

Results: 

I suggest to show descriptive statistics of the RE (phenotypes and all breeding values). Moreover, I would like to see spearm correlations of the animals ranking of phenotypes and breeding values (from different methodologies) across calving number. 

Please, improve the resolution and overall quality of the Figures 3 and 4.

 Table 5 - It seems odd to me. The correlations are too high. How is possible a correlation equals one between EGV3 and IpcT? Is it pearson correlation analysis? And spearman´s rank correlation is equally high, or not? Please, revise it carefully. 

Figure 7 - The title is not explaining well the content of the figures. 

Author Response

Thanks for your comments. We hope that this version of the manuscript is clearer and all your concerns are addressed.

The subject of the paper is very interesting. I suggest some changes in order to better explain the analysis of the data and improve the results presentation. Minor changes needed are highlighted in the pdf attached. 

First, I suggest to revise the title, which is too long and confusing. 

Thanks for the suggestion. We have modified the title following your advice.

Methodology: 

Line 129-130 - Is Herd-year-season related to the birth of the cow?

Is it possible to include the year/season of each breeding season in each observations? As the environment can be very different from one breeding season to another and it can affects the reproductive results. 

Thank you for your comment. We corrected the text for clarity L136-153. In the classical repeatability model (Rep) we used the combination: herd-year- season of birth of the cow (HYSN) as random effect. In random regression methodology (RA) we used the combination herd-year-breeding season (HYSk) in each observation.

Lines 131 to 134 - I looked at the reference and did not understand clearly the calculation of the RE. As this is essential for the paper, I suggest to add a clear explanation of how the calculation was done. I suggest to add description of the distribution of the data, what is the mean, median, etc... Seems to me that the numbers varies from 40% to 100%. Therefore, it is a censored data as cannot be higher than 100%. Moreover, I think is important to explain how many animals with information in each calving number. How the cow culling process affects the estimation of the breeding values? 

Thank you for your comment. Following your suggestions, the definition of RE and another reference has been added. Lines 83-89.

A table with descriptive statistics for input data has been added in line 261.

This trait evaluates the performance until the last calving of the cow, not until its culling, death or removal from the herd. Therefore, there aren’t censored data.

Line 199 - As IpcT comes from a summ from several estimatives, is it changing the distribution of the data? or even the mean, median, maximum and minimum? I suggest to add this descriptive statistics in results section. 

Thank you for your comment. As can be seen in Figure 6, there are some differences in the distribution. Information on the average, maximum and minimum of this parameter has been added. In the new variable there is a change of scale, so the statistics are not comparable. The correlation between the breeding values for Rep and IpcT shows a significant relationship but of a medium magnitude (0.45), which supports the use of this index as a pseudophenotype instead of the global vg of the animal for this character. L389-395.

Line 225 - 227 - Did you test not using the PCA correction? Because as you selected two contrasting groups of animals to compose the dataset, maybe you are loosing information of snps asssociated with the trait due this correction for population stratification. 

What breeding value was used for GWAS? Define it clearly. 

Thanks for the suggestion. Yes, we performed the GWAS without using the PCA correction. But we believe that it is important to correct for population structure because we must take into account the existence of strong family relationships into herd, common in livestock populations (van den Berg et al., 2019).

We used IpcT as pseudophenotype for GWAS. It is defined in the materials and methods section on line 201 to line 209.

Results: 

I suggest to show descriptive statistics of the RE (phenotypes and all breeding values). Moreover, I would like to see spearm correlations of the animals ranking of phenotypes and breeding values (from different methodologies) across calving number. 

Thank you for your comment. According to your suggestions, a descriptive statistics has been inserted L261. Table 6 shows the correlation between the breeding value at first calving and the IpcI, and the correlation between the breeding value and the different calvings are showed in Figure 4.

Please, improve the resolution and overall quality of the Figures 3 and 4.

Thank you for your comment. We have improved the resolution of figure 3, figure 4 has already the maximum resolution that the program allow.

 Table 5 - It seems odd to me. The correlations are too high. How is possible a correlation equals one between EGV3 and IpcT? Is it pearson correlation analysis? And spearman´s rank correlation is equally high, or not? Please, revise it carefully. 

Thank you for your comment. The Spearman correlations are practically identical except between the first calving and the rest after 5 calvings, which are slightly higher in the case of the Spearman connection.

np1

np2

np3

np4

np5

np6

np7

np8

np9

IpcT

np1

1,000

0,992

0,966

0,920

0,860

0,802

0,761

0,751

0,774

0,966

pc12

0,966

0,989

0,997

0,986

0,958

0,922

0,896

0,890

0,907

1,000

Figure 7 - The title is not explaining well the content of the figures. 

Thanks for the suggestion. We have clarified the title.

Reviewer 2 Report

The manuscript presents an analysis of reproductive efficiency in Retinta cattle using a repeatability and random regression model.

Line 31: reproductive efficiency behavior is not homogeneous throughout the cow’s life. >> please rephrase. Not clear whether you refer to phenotypic or genetic variation.

reproductive efficiency behavior >> reproductive efficiency

Line 33-34: cn is not explained.

Line 80 : reproductive efficiency: please outline the definition of RE. Use authors always the same definition.

Line 81: other reproductive traits in several livestock species: not explained. Remains unclear. References are missing.

Line 76-90: more specific information on RE would be desired.

Line 132-134: please explain what you are calculating to get RE

How do you treat animals who left the herd unvoluntarily.

Have you records for the dates when cows are leaving the herd.

Line 192-201: not very easy to follow what authors are doing. Also confusing are the number of calvings (6+, 9 or 10).

Line 199: rationale behind IpcT has to be explained.

Line 213: dataset was pruned by linkage disequilibrium in PLINK using the –indep-pairwise 50 5 0.5: do not understand why you prune the genotype data. You may miss some associated SNPs.

Line 219: The model equation is confusing. Please amend.

Line 220: W is an incidence matrix of covariates (fixed effects) including a column of 1 s; Which fixed effect did you include, just a general mean.

Figure 1: hard to understand because a clear definition of RE is missing.

Younger should also reach values >80% if they reproduce with short intercalving intervals.  CI and range of RE should be given.

Figure 2: how did you estimate this effect for birth years. Did you use the additive genetic effect of the cow?

Author Response

Thanks for your comments. We hope that this version of the manuscript is clearer, and all your concerns are addressed.

The manuscript presents an analysis of reproductive efficiency in Retinta cattle using a repeatability and random regression model.

Line 31: reproductive efficiency behavior is not homogeneous throughout the cow’s life. >> please rephrase. Not clear whether you refer to phenotypic or genetic variation.

reproductive efficiency behavior >> reproductive efficiency

Thank you for your comment. We have rephrased it.

Line 33-34: cn is not explained.

Thank you for your comment. We have explained it.

Line 80 : reproductive efficiency: please outline the definition of RE. Use authors always the same definition.

The reviewer is right. The definition of RE and another reference have been added. L83-86

Line 81: other reproductive traits in several livestock species: not explained. Remains unclear. References are missing.

Thank you for your comment. We have explained it. L88

Line 76-90: more specific information on RE would be desired.

The reviewer is right. The definition of RE and another reference have been added. L83-86.

Line 132-134: please explain what you are calculating to get RE

How do you treat animals who left the herd unvoluntarily.

Have you records for the dates when cows are leaving the herd.

The reviewer is right. The definition of RE and another reference has been added. Lines 83-86.

This trait evaluates the performance until the last calving of the cow, not until its culling, death or removal from the herd. Therefore, there aren’t censored data.

Line 192-201: not very easy to follow what authors are doing. Also confusing are the number of calvings (6+, 9 or 10).

Thank you for your comment. In the classical repeatability model (Rep) an effect of the total number of calvings fo the cow was added, grouped into 6 classes. While In random regression methodology (RA), each of the calvings was analyzed individually from the first to the 9th. The number 10 was a erratum, we corrected it to 9.

Line 199: rationale behind IpcT has to be explained.

Thank you for your comment. The connection between the breeding values for the classical repeatability model (Rep) and IpcT shows a significant relationship but of a medium magnitude (0.45), which supports the use of this index as a pseudophenotype instead of the global vg of the animal for this trait. We have completed the text with this information. L392-395

Line 213: dataset was pruned by linkage disequilibrium in PLINK using the –indep-pairwise 50 5 0.5: do not understand why you prune the genotype data. You may miss some associated SNPs.

Thank you for your comment. Based on other GWAS analyses, we believe it is necessary to eliminate SNPs with strong linkage disequilibrium in these analyses.

Line 219: The model equation is confusing. Please amend.

Thank you for your comment. But it is the usual way of expressing the GWAS models analyzes with GEMMA.

Line 220: W is an incidence matrix of covariates (fixed effects) including a column of 1 s; Which fixed effect did you include, just a general mean.

Thank you for your comment. We included as fixed effects the IpcT and a general mean. The general average is added because it is recommended by the author of the program.

Figure 1: hard to understand because a clear definition of RE is missing.

Younger should also reach values >80% if they reproduce with short intercalving intervals.  CI and range of RE should be given.

Thanks for the suggestion. The definition of RE and another reference have been added.

In younger cows whose age at first calving is far from optimal, the initial values of reproductive efficiency are lower. These values can be later compensated with short calving intervals an become higher.

Figure 2: how did you estimate this effect for birth years. Did you use the additive genetic effect of the cow?

Thank you for your comment. It was calculated with the genetic values average of the cows born in each year.

Reviewer 3 Report

General comments:

In this manuscript, authors mainly analyzed reproduction efficiency (RE) of Retinta Spanish cattle breed using classical repeatability and random regression models. The novelty and validity of this study in that it used a large number of individuals and addressed fertility traits from a new perspective is highly commendable. The manuscript is well written and the results are clearly presented. I have a few minor comments, explained below.

Specific comments:

L223: “-1” should be a superscript.

L431: Please provide the rationale for setting the thresholds.

Author Response

Thanks for your comments. We hope that this version of the manuscript is clearer and all your concerns are addressed.

General comments:

In this manuscript, authors mainly analyzed reproduction efficiency (RE) of Retinta Spanish cattle breed using classical repeatability and random regression models. The novelty and validity of this study in that it used a large number of individuals and addressed fertility traits from a new perspective is highly commendable. The manuscript is well written and the results are clearly presented. I have a few minor comments, explained below.

Specific comments:

L223: “-1” should be a superscript.

Thanks. You are right. We have modified it.

L431: Please provide the rationale for setting the thresholds.

 Thank you for your comment. According to the GWAS studies performed on livestock, we believe there is no clear or exact answer about setting the threshold. The threshold is usually set in the range of 10-3 to 10-6.

Following the analysis of Wang, et al. 2018 about GWAS study for reproductive traits in swine, we decided to set the threshold at 10-4, as we believe it is a preliminary analysis that needs to be completed with a larger number of samples.
